# New Water-Soluble Poly(propylene imine) Dendrimer Modified with 4-Sulfo-1,8-naphthalimide Units: Sensing Properties and Logic Gates Mimicking

**DOI:** 10.3390/s23115268

**Published:** 2023-06-01

**Authors:** Awad I. Said, Desislava Staneva, Ivo Grabchev

**Affiliations:** 1Faculty of Medicine, Sofia University “St. Kliment Ohridski”, 1407 Sofia, Bulgaria; 2Department of Chemistry, Faculty of Science, Assiut University, Assiut 71516, Egypt; 3Department of Textile, Leader, and Fuels, University of Chemical Technology and Metallurgy, 1756 Sofia, Bulgaria; grabcheva@mail.bg

**Keywords:** poly(propylene imine) dendrimer, 1,8-naphthalimides, aggregation-induced emission (AIE), excimer, solvatochromism, water purity, INHIBIT, XNOR, digital comparator

## Abstract

A new water-soluble poly(propylene imine) dendrimer (PPI) modified with 4-sulfo-1,8-naphthalimid units (**SNID**) and its related structure monomer analog (**SNIM**) has been prepared by a simple synthesis. The aqueous solution of the monomer exhibited aggregation-induced emission (AIE) at 395 nm, while the dendrimer emitted at 470 nm due to an excimer formation beside the AIE at 395 nm. Fluorescence emission of the aqueous solution of either **SNIM** or **SNID** was significantly affected by traces of different miscible organic solvents, and the limits of detection were found to be less than 0.05% (*v*/*v*). Moreover, **SNID** exhibited the function to execute molecular size-based logic gates where it mimics XNOR and INHIBIT logic gates using water and ethanol as inputs and the AIE/excimer emissions as outputs. Hence, the concomitant execution of both XNOR and INHIBIT enables **SNID** to mimic digital comparators.

## 1. Introduction

Dendrimers are three-dimensional star-shaped supramolecular architectures having various functional groups in the structure. They have recently been attracting the attention of scientists as an alternative to linear and branched polymers due to their unique structural features, including large surface area and the flexibility to incorporate different compounds into their periphery or interior parts [1,2,3]. Over the last two decades, numerous structural scaffolds for dendrimers have been reported, ranging from pure organic molecular frameworks to organometallic and biomaterials [4,5,6]. Exploration of dendrimers applications in supramolecular chemistry is still ongoing. Recently, many reports have presented their potential in drug delivery [7], tissue engineering [8], bio-imaging [9], catalysis [10], cancer therapy [11], and a variety of other applications. Luminescent dendrimers are indispensable components in high-technology industries, particularly optoelectronics, light-harvesting antennae in solar cells, sensors for detecting pollutants in the environment, biology, and medicine [12,13,14].

The peripheral functionalization of dendrimers with different chromophores such as dansyl sulfonate [15,16], pyrene [17], azobenzenes [18], and coumarin [19,20] moieties and their potential applications have been described. Our previous works have been focused on the functionalization of poly(amidoamine) [21,22,23,24,25,26,27,28,29,30] and poly(propylene amine) [31,32,33,34] dendrimers by 1,8-naphthalimide chromophor groups and studied their potential as sensors for transition metal cations and pH. 1,8-Naphthalimides are promising fluorophores for designing fluorescent sensors because of their good photostability, strong fluorescence emission, high quantum yield, and flexibility to be modified. Molecular architectures based on 1,8-naphthalimides are well-known assemblies with bright emissive color, high photostability, and sensor activities for colorimetric and fluorometric probing [35,36,37,38,39].

Aggregation-based luminogens are non-emissive molecules that become highly emissive in solution by limiting their intramolecular rotation (RIR) in the aggregated state [40]. There are many reports about the applications of AIEgens, including liquid crystals [41], organic light-emitting diodes (OLED) [42], photoluminescent agents [43], and sensors [44]. Moreover, aggregation makes some organic fluorophores exhibit a new red-shifted emission caused by excited dimers, excimers, formed by associating two fluorophore units when they are a close vicinity to each other [45]. There are two kinds of excimers (i) dynamic excimers [46] resulting from associating a fluorophore in the excited state and (ii) static excimers that are formed in the ground state [47]. These excimers absorb like monomers, but the emission is red-shifted related to monomer emission [48,49]. The excimer emission is extremely sensitive to the polarity of the solvent [50].

Water contamination by organic solvents is disadvantageous for the progress of chemical reactions, biological processes, pharmaceuticals, and foodstuffs production [51,52,53,54]. Traditional methods for detecting traces of water in organic solvents, such as analytical, chromatographic, and electrochemical methods, suffer from many drawbacks like toxic agents, expensive instruments, and complicated operations [55,56,57]. Recently, many organic molecular sensors for detecting water pollution have been reported, though they require multistep synthesis, and the sensitivity achieved is low [58,59,60].

Recently, there has been considerable progress in developing optical molecular sensing systems to mimic logic gates and operations for incorporation into information technology instead of silicon-based ones [61,62,63,64,65]. A digital comparator to compare two inputs can be constructed by the combinational logic circuit of three INHIBIT logic gates [66,67] or of XNOR/INHIBIT gates [68,69]. 

In this work, a novel water-soluble PPA dendrimer modified with 4-sulfo-1,8-naphthalimides units was synthesised as a part of our ongoing research on the synthesis and characterization of novel periphery functionalized with 1,8-naphthalimides dendrimers. The monomer of the dendrimer has also been synthesized and examined so that the results from the investigations of the photophysical properties, solvatchromism, and sensory function of both the monomer and the dendrimer could be compared. Experiments on the excimer formation induced by the aggregation of the dendrimer were carried out as well. Moreover, the function of both compounds to mimic logic gates was studied. 

## 2. Materials and Methods

The first generation (poly propylene imine) dendrimer (PPI), 4-Sulfo-1,8-naphthalic anhydride potassium salt and *N,N-*dimethyltrimethylenediamine were purchased from Sigma Aldrich and used without purification. All used solvents (Sigma Aldrich, St. Louis, MO, USA): dimethylsulfoxide (DMSO), *N,N*-dimetjylformamide (DMF), tetrahydrofuran (THF), dichloromethane (DCM), ethanol, dioxane were of spectroscopic grade purity. ^1^H and ^13^C-NMR spectra were recorded at ambient temperature in DMSO-d_6_ as a solvent on a Bruker Avance II+ 600 spectrometer operating at 600.13 MHz and 151 MHz, respectively. The UV-Vis absorption and emission spectra were recorded on Varian Cary 5000 UV-Vis-NIR spectrophotometer and on a “Cary Eclipse” spectrofluorometer, respectively, using 1 cm optical path length quartz cuvettes (Hellma, Müllheim im Markgräflerland, Germany). Slits width of 5 nm for the excitation and emission. All of the measurements were measured at 25.0 °C. TLC monitoring was conducted using silica gel (Fluka F_60_ 254 20 × 20; 0.2 mm) and toluene/methanol/ (4:1) as an eluent. OriginPro 8 software for data processing has been used. Stock solutions of SNIM and SNID were prepared in DMF as 10^−2^ M to ensure negligible volumes of the stock to reach the required concentration (3 μL for 10^−5^ M and 1.5 μL for 5 × 10^−6^ M) using 3 mL as a total volume of the solvent(s). 

### 2.1. Synthesis of Potassium 2-(3-(Dimethylamino)propyl)-1,3-dioxo-2,3-dihydro-1H-benzo[de]isoquinoline-6-sulfonate SNIM

*N,N-*Dimethyltrimethylenediamine (250 μL, 2 mmole) were added dropwise to a suspension of 4-sulfo-1,8-naphthalic anhydride **1** (0.53 gm, 1.7 mmole) in 25 mL of ethanol and was refluxed for 4 h. The final product was isolated after filtration of the solid and washing with ethanol. Yield 95%, 0.61 g, m.p. > 300 °C. 

FT-IR (KBr) cm^−1^: 3080 (νCH (Aromatic)); 2960, 2860, 2810, 2780 (νCH (Aliphatic)); 1700, 1650 (νC=O). ^1^H NMR (600 MHz, DMSO) δ 9.24 (dd, *J* = 8.6, 1.1 Hz, 1H), 8.49 (dd, *J* = 7.2, 1.1 Hz, 1H), 8.46 (d, *J* = 7.5 Hz, 1H), 8.21 (d, *J* = 7.5 Hz, 1H), 7.88 (dd, *J* = 8.6, 7.3 Hz, 1H), 4.09–4.04 (m, 2H), 2.31 (t, *J* = 7.0 Hz, 2H), 2.12 (s, 6H), 1.80–1.72 (m, 2H). ^13^C NMR (151 MHz, DMSO) δ 164.1, 163.6, 150.2, 134.5, 130.8, 130.6, 128.6, 128.0, 127.3, 125.4, 123.2, 122.6, 57.2, 45.5, 38.7, 25.9. Analysis: C_17_H_17_N_2_O_5_K S (400.22 g mol^−1^): Calc. (%): C-46.60, H 4.40, N 7.25; Found (%): C-46.83, H 4.44, N 7.31.

### 2.2. Synthesis of 4-Sulfo-1.8-naphalimide Based PPI Dendrimer SNID

The poly(propylene imine) dendrimer from first generation (0.32 g, 1 mmol) and 4-sulfo-1,8-naphthalic anhydride **1** (1.3 g, 4 mmol) were refluxed in 25 mL ethanol, and the reaction progress has been monitored by TLC. After 4 h, the product was filtered, washed with ethanol, and dried. Yield: 1.34 g (98%), decomposed at temperatures higher than 300 °C.

FT-IR (KBr) cm^−1^: 3070 (νCH (Aromatic)); 2950, 2850, 2810 (νCH (Aliphatic)); 1695, 1651 (νC=O). ^1^H NMR (600 MHz, DMSO) δ 9.21 (dd, *J* = 8.6, 1.2 Hz, 4H, Ar-H), 8.43 (d, *J* = 7.5 Hz, 4H, Ar-H), 8.38 (d, *J* = 7.6 Hz, 4H, Ar-H), 8.20 (d, *J* = 7.6 Hz, 4H, Ar-H), 7.81 (dd, *J* = 8.6, 7.3 Hz, 4H, Ar-H), 4.13–4.03 (m, 8H, (OC)_2_N*CH*_2_), 3.10–2.80 (m, 4H, *CH*_2_N<), 2.60–2.55 (m, 8H, *CH_2_*N(CO)_2_), 1.83–1.74 (m, 8H, (OC)_2_NCH_2_*CH*_2_CH_2_N), 1.51–1.43 (m, 4H, >CH_2_*CH*_2_*CH*_2_CH_2_N<). ^13^C NMR (151 MHz, DMSO) δ 164.0 (C=O), 163.6 (C=O), 150.0, 134.4, 130.7, 130.49, 128.5, 127.9, 127.2, 125.5, 123.3, 122.5 (10 Ar. C), 51.4, 38.9, 25.4 (aliph C). Analysis: C_64_H_52_N_6_O_20_K_4_S_4_ (1509.15 g mol^−1^): Calc. (%): C-50.89, H 3.45, N 5.57; Found (%): C-50.80, H 3.49, N 5.52.

## 3. Results and Discussion

### 3.1. Design and Synthesis of the Probe

The synthesis of 4-sulfo-1,8-naphthalimide-modified PPA dendrimer **SNID** and its related monomer **SNIM** is presented in Figure 1. Their chemical structures were confirmed by UV-Vis absorption, fluorescent, FT-IR, and NMR spectra (Appendix A). The π-π stacking of 1,8-naphthalimide units is favoured in water, and hence, aggregation-induced emission AIE is possible. The function of the dendrimer scaffold is to stick close to the 4-sulfo-1,8-naphthalimde moieties, thus enabling the aggregation-induced excimer formation in water. 

### 3.2. Photophysical Characteristics

The influence of solvent polarity on the absorption and emission spectra of **SNIM** and **SNID** has been investigated, and the respective data have been summarized in Table 1. The absorption spectra of monomer and dendrimer have an absorption band in the range of 300–370 nm corresponding to the 4-sulfo-1,8-naphthalimide chromophore group, Figure 1. While the absorption band was characterized by a well-developed vibrational fine structure in most solvents, the structure is almost blurred in the hydroxylic solvents due to hydrogen bonding with the solvent molecules and π–π stacking that restricts the vibrational transitions. The position of the absorption band is not affected significantly by the solvent polarity, suggesting that these compounds in the ground state are not sensitive to the polarity of the environment. The solvent polarity has an impact only on the vibrational transitions. On the other hand, the absorption spectra of the dendrimer **SNID** in different solvents are similar to the ones of the monomer **SNIM**, except the molar extinction coefficients at the absorption maxima, which are approximately four times higher than those of the monomer **SNIM,** which indicates the full substitution of the primary amino groups in the dendrimer periphery by 4-sulfo-1,8-naphthalimide units [70]. 

Regarding fluorescence emission, after excitation at 340 nm, the monomer gives a strong fluorescence emission centred at 392 nm only in water, Figure 2A. It is attributed to the monomer molecules aggregation that is induced by the π–π stacking. This stacking restricts the nonradiative vibrational de-excitations processes of the excited molecules. Moreover, the monomer **SNIM** gives a weak emission in ethanol and DCM due to the vague formation of aggregates in these solvents. The fluorescence emission observed in THF, despite the well-developed vibrational fine structure of the absorption band, refers to the aggregation favoured in the excited state rather than in the ground state. Strikingly, the behaviour of the dendrimer in water is different from that of the monomer, where besides the emission at 395 nm, which is weaker, a strong emission centred at 475 nm is observed (Figure 2B). This is confirmed by the photograph of the CNID and SNIM compounds dissolved in water, DMF, and ethanol and irradiated with monochromatic UV light at 366 nm. The figure shows the blue-green fluorescence emission of CNID in an aqueous solution, while SNIM emits blue fluorescence (Figure 2C). The former emission, as mentioned above, is caused by the excimer formation of 4-sulfo-1,8-naphthalimide units, while that of the latter is due to the aggregation of dendrimer molecules [71]. 

The discriminated fluorescence emission of the monomer and its dendrimer in water encouraged us to investigate the applicability of these compounds as probes for quantitative measurements of the purity of water contaminated with another miscible organic solvent. We used ethanol, DMF, and dioxane as representatives for polar protic and aprotic and nonpolar solvents, respectively. Moreover, we investigated the influence of water traces in the solvents on the emission response of the **SNIM** and **SNID**. 

### 3.3. Solvatochromism of SNIM

It has been found that ethanol has no effect on the emission of the aqueous solution of **SNIM** till 60% (*v*/*v*) of ethanol. Higher amounts of ethanol (>60%) led to emission quenching at 395 nm due to the dissociation of aggregates by ethanol molecules. On the other hand, fluorescence emission at 395 nm of ethanol solution of **SNIM** has enhanced by adding water, Figure 3. The limit of detection (LOD) for water presence in ethanol was found to be 0.09% by volume. LOD was calculated using LOD = 3σ/b [38], where *b* is the slop and *σ* is the standard deviation. The increase in the emission by adding water is ascribed to the aggregation of **SNIM** molecules induced by π–π stacking of nonpolar 1,8-naphthalimide moieties in the presence of water. The low LOD of **SNIM** towards water presence in ethanol indicates that it can be used as a low-cost reagent for the detection of traces of water in alcohol. The required volume of water to reach saturation of the fluorescence response of **SNIM** in the ethanol solution was found to be ≈24% (*v*/*v*) (Appendix A). 

Moreover, **SNIM** exhibited the ability to investigate the contamination of water in DMF, as a representative for polar aprotic solvents, by its fluorescence emission, Figure 4. Similar to ethanol, the presence of DMF decreased the emission of SNIM in water due to the dissociation of aggregated molecules by DMF solvation. The limit of detecting DMF contamination was found to be 0.08%, refereeing to the applicability of **SNIM** to detect traces of DMF in water. The saturation of emission response was reached after the addition of 5% of DMF to the water solution, after which the decrease in fluorescence with increasing DMF content up to 10% is negligible (Figure 4B).

Moreover, the effect of dioxane, as a representative of nonpolar solvents, on the emission of a **SNIM** solution in water has also been investigated (Figure 5). In this case, the emission is quenched by the presence of dioxane traces due to the dissociation of π–π stacking between 1,8-naphthalimide moieties. The LOD and dioxane volume required to reach saturation was found to be 0.05% and 10%, respectively. Moreover, the influence of water presence on the emission of **SNIM** solution in dioxane has been investigated. Contrarily, the presence of water traces enhanced the fluorescence emission. The limit of detection of water in dioxane was found to be 0.14%. Hence, **SNIM** has a dual sensitive sensory applicability for investigating the purity of both water and dioxane in the presence of the other as a contaminant. In other words, **SNIM** is able to detect the presence of dioxane traces in a water sample and water traces in a dioxane sample.

### 3.4. Solvatochromism of Dendrimer SNID

The effect of water traces on the emission of dendrimer solution was examined in ethanol solution. As shown in Figure 6, water leads to emissions enhancement at both 395 nm and 470 nm, and the limits of detection were found to be 0.5% and 1%, respectively. The fluorescence enhancement at 395 nm was observed till 50% water fraction; after that, the emission quenched by further water addition, Figure 7, due to the higher rate of excimer formation and to the fact that more 1,8-naphthalimides unites become included in the excimer formation. In concomitance, the emission at 470 nm increased slowly till 50 % water fraction, then further addition of water increased the rate. Behaviour of the dendrimer in the presence of both water and ethanol solutions as inputs and the emission at 395 nm (λ_ex._ = 340 nm) as output and using the initial case of 50% water fraction mimics XNOR logic gate, Figure 7C, where at the initial state (water coded as 0 and ethanol as 0), the output is high (coded as 1). Addition of ethanol till water fraction = 20% (ethanol coded as 1 and water coded as 0) gets the emission at 395 nm low (coded as 0). Moreover, the addition of water till it reaches a water fraction of 80 % (ethanol coded as 0 and water coded as 1) gets the emission low and coded as 0. Finally, the addition of both ethanol and water in equal amounts (both coded as 1) retains the initial state (emission gets high and coded as 1). On the other hand, using the emission at 470 nm as output and the emission threshold shown in Figure 7B, **SNID** mimics INHIBIT logic gate where the emission can be considered high (coded as 1) only in the case of adding water alone and otherwise the emission is low (coded as 0). Moreover, a combination of XNOR and INHIBIT logic gates works as a digital comparator, Figure 7D.

Furthermore, the applicability of the dendrimer **SNID** for detecting DMF contamination in water has been studied, Figure 8. The addition of DMF traces to **SNID** solution in water was associated with quenching the emissions at 395 nm and 470. The limits of detection were found to be 0.09% and 0.2% using emissions at 395 nm and 470 nm, respectively. The quenching of the emissions by DMF contamination is linear in the range of 0–1% of a DMF fraction. On the other hand, the emission spectrum of **SNID** solution in DMF was affected only by large volumes of water, Figure 9, due to the good solvation of DMF to dendrimer molecules. 

Moreover, the applicability of **SNID** for detecting dioxane contamination in water samples has been investigated. The presence of dioxane traces quenched both the emissions at 395 and 470 nm linearly in the range of 0–2%. The LOD values were found to be 0.3% and 0.7% using emissions at 395 nm and 470 nm, respectively, Figure 10. On the other hand, the addition of water to **SNID** solution in dioxane is associated with enhancing the emissions at 395 nm and 470 nm, like the case of adding water to **SNID** solution in ethanol, Figure 11. Hence, **SNID** can act as a digital comparator using water and dioxane as inputs and the emissions at 395 nm and 470 nm as outputs.

## 4. Conclusions

This work presents the synthesis of a new water-soluble poly(propylene amine) dendrimer from the first generation, modified with 4-sulfo-1,8-naphthalimide **SNID** and its monomer analog **SNIM** for detecting water contamination by different organic solvents. Both the monomer and dendrimer aggregate in the aqueous solution because of the π-π stacking of 4-sulfo-1,8-naphthalimide moieties that allows the formation of excimers between the excited and non-excited 4-sulfo-1,8-naphthalimide fragments of dendrimer molecules. Moreover, the incorporation of 4-sulfo-1,8-naphthalimide units into the dendrimer scaffold improves their tolerance towards strong bases. Furthermore, the dependence of the emissions, caused by the aggregation and excimer formations, on water presence enables these molecules to detect the presence of traces of various organic solvents in water and vice versa. It has been shown that the dendrimer **SNID** mimics both XNOR and INHIBIT logic gates which work in combination to execute the function of the digital comparator.

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
