# Peer review of "New Water-Soluble Poly(propylene imine) Dendrimer Modified with 4-Sulfo-1,8-naphthalimide Units: Sensing Properties and Logic Gates Mimicking"

_sensors, 2023, doi:10.3390/s23115268_

Round 1

Reviewer 1 Report

Authors describe in this work, the synthesis of a novel water soluble PPA dendrimer modified with 4-sulfo-1,8-naphthalimides units together with their corresponding monomer for detecting water contamination by using different organic solvents. Subsequently, dendrimer and the monomer of the dendrimer were characterized in terms of their photophysical properties, solvatochromism and sensory function. Furthermore, authors have performed experiments on the excimer formation induced by the aggregation of the dendrimer. Moreover, the function of both compounds to mimic logic gates was studied.

This paper is well written, clear, and well structured. And although it is concise in relation to the subject matter and well explained, there are some little questions that should be addressed. Therefore, I recommend this article for publication with a minor revision:

 Figure captions should be improved with more explanatory data.

The authors should explain in more detail the experimental part performed and the way of performing the described measurements.

In keywords, there is a mistake in word “solvatochromism”.

In line 85, “dichloromethane”

In line 183, figure S9 is missing.

In line 190, the authors should explain figure 4B more clearly. It is somewhat confusing and not well understood since the previous figure refers to water contamination in ethanol.

Author Response

  1. Figure captions should be improved with more explanatory data.

 The captions have been corrected

  1. The authors should explain in more detail the experimental part performed and the way of performing the described measurements.

 Some additional data have been added to the experimental part.

  1. In keywords, there is a mistake in word “solvatochromism”.

It was corrected

  1. In line 85, “dichloromethane”

It was corrected

  1. In line 183, figure S9 is missing.

Figure S9 has been changed with S7 at added to the supplementary  information.

  1. In line 190, the authors should explain figure 4B more clearly. It is somewhat confusing and not well understood since the previous figure refers to water contamination in ethanol.

The explanation has been done.

Reviewer 2 Report

In the manuscript entitled, ‘New Water Soluble Poly(propylene imine) Dendrimer modified with 4-sulfo-1,8-naphthalimide units: Sensing properties and Logic Gates Mimicking ’ It expands the application of dendritic polymers in aqueous environment detection and provides a good attempt of dendritic polymers in environmental detection, but there are still some problems, and my suggestion is to publish it after the major revision.

1) According to figure 2 A&B, SNIM and SNID molecules have a clear distinction in the occurrence of peaks, can their photos be provided to illustrate?

2) The horizontal axis interval ranges in Figure 1 are not uniform, and the same problem appears in Figure 2.

3) Please provide a high resolution image of Figure 7E.

4) Can macroscopic photographs be provided in support of the variation of emission peak intensity of SNIM and SNID molecular solutions for different solvent detection?

5) The volume change arrow of DMF solvent appears twice in Figure 8A, please check.

6) In the supporting information, in the IR spectra of Figure S3 and S6, please correct the figure notes E2 and E1 to the target molecule, and can you label the functional group peaks.

7) Can you provide mass spectra for SNIM and SNID.

Author Response

According to figure 2 A&B, SNIM and SNID molecules have a clear distinction in the occurrence of peaks, can their photos be provided to illustrate?

The macrograph of bot compounds in water, DMF, and ethanol solutions was added to the text (Figure 2C)

2) The horizontal axis interval ranges in Figure 1 are not uniform, and the same problem appears in Figure 2.

Both figures have been changed

3) Please provide a high resolution image of Figure 7E.

The resolution in Figure 7E has been improved.

4) Can macroscopic photographs be provided in support of the variation of emission peak intensity of SNIM and SNID molecular solutions for different solvent detection?

5) The volume change arrow of DMF solvent appears twice in Figure 8A, please check.

 Figure 8A has been changed

6) In the supporting information, in the IR spectra of Figure S3 and S6, please correct the figure notes E2 and E1 to the target molecule, and can you label the functional group peaks.

The notes of Figures S3 and S6 have been corrected. Unfortunately, the peaks in the functional groups cannot be labeled. Their exact values have been determined by plotting the spectra with the program Origin.

7) Can you provide mass spectra for SNIM and SNID.

At this stage, we cannot conduct mass spectrometric analysis. We used elemental analysis to determine the purity of the new compound. 

Round 2

Reviewer 2 Report

The authors have revised the manuscrpt according to some advices, so it should be recommended for publication.